# Spine-like Joint Link Mechanism to Design Wearable Assistive Devices

**DOI:** 10.3390/s22062314

**Published:** 2022-03-17

**Authors:** Jung-Yeong Kim, Jung-San Cho, Jin-Hyeon Kim, Jin-Tak Kim, Sang-Chul Han, Sang-Shin Park, Han-Ul Yoon

**Affiliations:** 1Robotics, University of Science and Technology (UST), Daejon 34113, Korea; kjy880527@kitech.re.kr (J.-Y.K.); chojs@kitech.re.kr (J.-S.C.); 2Robotics R&D Department, Korea Institute of Industrial Technology (KITECH), Ansan 15588, Korea; qkrb0117@kitech.re.kr (J.-H.K.); jintagi@kitech.re.kr (J.-T.K.); bomber21@kitech.re.kr (S.-C.H.); pss@kitech.re.kr (S.-S.P.); 3Division of Software, Yonsei University, Wonju 26493, Korea

**Keywords:** spine-like joint link mechanism, wearable assistive device, exoskeleton component, thoracic/lumbar spine assistive device

## Abstract

When we develop wearable assistive devices, comfort and support are two main issues that need to be considered. In conventional design approaches, the degree of freedom of the wearer’s joint movements tends to be oversimplified. Accordingly, the wearer’s motion becomes restrained and bone/ligament injuries might occur in case of an unexpected fall. To mitigate these issues, this paper proposes a novel joint link mechanism inspired by a human spine structure as well as functionalities. The key feature of the proposed spine-like joint link mechanism is that hemispherical blocks are concatenated via flexible synthetic fiber lines so that their concatenation stiffness can be adjusted according to a tensile force. This feature has a great potentiality for designing a wearable assistive device that can support aged people’s sit-to-stand action or augment spinal motion by regulating the concatenation stiffness. In addition, the concatenated hemispherical blocks enable the wearer to move his/her joint with full freedom, which in turn increases the wearer’s mobility and prevents joint misalignment. The experimental results with a testbed and a pilot wearer substantiated that the spine-like joint link mechanism can serve as a key component in the design of wearable assistive devices for better mobility.

## 1. Introduction

Wearable assistive robots have been used for various applications, e.g., quiet balance support for elderly people [1,2,3], gait function rehabilitation for stroke survivors [4,5,6], heavy object lifting for manual workers [7,8,9], mobility augmentation for soldiers in challenging battle fields [10,11,12], and so on. Since the wearable assistive robots have shown great potential in a broad range of applications, many studies are now underway to extend their usage as human assistive devices.

At the early stage of wearable assistive robots, design approaches tended to simplify human joint-link mechanisms. For example, a knee joint was typically simplified as a hinge-type joint with a single degree of freedom (DoF), which in turn restricted thigh and shin movement only into the sagittal plane [13,14,15,16]. Moreover, most of the wearable assistive robots have been made by rigid materials, although they targeted the support of various body parts such as lower-limb [13,14,15,16,17,18,19,20], lower-back [21,22], upper-limb [23,24], and hand [25,26]. The rigidity of the wearable assistive robots might facilitate “enough support” in the sense of force/torque generation, whereas the length of a link could not be changed. Consequently, in addition to the aforesaid sagittal plane restriction, the wearer’s joint-link movement was constrained by a loss of flexibility in length.

The wearer’s constrained joint-link movements are well-exemplified by existing early-designed exoskeletons. Lower-limb exoskeletons such as EXPOS [13], BLEEX [14], HAL [15], eLEGS [16] commonly employed a uniaxial revolute joint model for a wearer’s knee joint, which was simplified to a 1-DoF model. Studies in [18,19,20,21] presented exoskeletons to support a wearer’s lower-back as well as lower-limb. The presented exoskeletons could retain the wearer’s lower-back in an upright position by supporting the lumbar spine with a rigid structure and the wearer’s knee joint was simplified as a hinge-type joint. Under all the constrained design approaches above, kinematic discrepancy between the wearer’s joint and the exoskeleton’s joint movements becomes inevitable.

The kinematic discrepancy between a human wearer and an exoskeleton robot indeed causes somewhat problematic issues. Even though the ranges of motion of hip/knee/ankle joints are rather limited, they are able to move in the medial–lateral direction as well as the anterior–posterior direction. Namely, the hip/knee/ankle joints are not perfect spherical joints, but their movements are still closer to those of the spherical joints [27,28]. Therefore, the kinematic discrepancy can cause unnatural walking and discomfort in the wearer during daily activities. Moreover, the reduced DoF due to the simplified kinematics exposes the wearer to serious joint or ligament injuries when the wearer tries bending/leaning toward the medial–lateral direction [28].

Wearable robots made by soft materials have been thought to be a feasible solution to alleviate both kinematic discrepancy and joint misalignment problems. Accordingly, various soft-type wearable robots have been developed, e.g., exo-suit [10], exo-glove [29], soft power suit [30], inflatable soft wearable robot [31,32], and so on. Compared to the rigid-type wearable robots, the soft-type ones constrain natural joint movement less, which in turn reduces discomfort caused by joint misalignment. The limitation of the soft-type wearable robots is, however, that they generate less assistive force/torque than the rigid-type. For instance, the soft-type wearable robots encourage a 10 to 100 percent increment in original force/torque generated by the wearer’s muscle; by contrast, the rigid-type devices produce a 100 to 900 percent increment [33]. Due to this limitation, the soft-type devices have been narrowly applied, e.g., to wrist support, forearm assistance, ankle support, etc.

To mitigate the above-mentioned issues of rigid/soft-type wearable robots, this paper proposes a novel joint link mechanism inspired by the dexterous characteristics of a human spine. Specifically, the distinguished feature of the proposed mechanism is that multiple hemispherical blocks are concatenated through synthetic fiber lines and rubber strips to functionally mimic the vertebrae, tendons, and ligaments of the spine; accordingly, it is referred to in this paper as a spine-like joint link mechanism (SJLM). The primary purpose of this study is to design and test the feasibility and efficacy of a novel joint-link mechanism mimicking the human spine. This study was initiated for the ultimate purpose of developing wearable assistive devices, for which the proposed SJLM is to serve as a key component. Throughout this paper, therefore, we mainly focus on explaining the design and fabrication of the SJLM, and feasible applications are discussed as well.

This paper is organized as follows. In Section 2, the design of the SJLM is presented. Specifically, component design, fabrication, and dexterous functionalities are explained. Section 3 discusses the input–output relationship of the SJLM, which corresponds to a tensile force under a hydraulic actuation versus an assistive force generated at an end-link part. Section 4 presents the experimental procedures and results to substantiate the input–output relationship and the maximum supporting capabilities. Two feasible SJLM applications are introduced as well. Section 5 is the conclusion of this paper.

## 2. Design of Spine-like Joint Link Mechanism

### 2.1. Structure Overview

Figure 1 illustrates a SJLM design inspired by a human spine and functional correspondences and presents the fabricated SJLM. Figure 1a shows bones, muscles and tendons, ligaments, and cartilages (which are also referred to as vertebral body), erector spinae, anterior/posterior longitudinal ligaments, and intervertebral discs in terms of anatomic jargon, respectively. Inspired by this anatomical structure and these functionalities, as presented in Figure 1b, the SJLM was designed with hemispherical blocks, a hydraulic actuator, synthetic fiber lines, a silicon-based strip, and natural rubber discs which functionally correspond to bones, muscles and tendons, ligaments, and cartilages, respectively. Figure 1c shows the fabricated SJLM: isometric view, front view, and side view.

The detailed specification of the SJLM along with its hydraulic cylinder is described in Table 1. We note that the specification presented in Table 1 is an exemplification of the SJLM design and that it can be modified flexibly according to the wearer’s body measurement as well as design materials.

### 2.2. Components Design and Fabrication

Recall that the components of the SJLM and their functional correspondences were presented in Figure 1b; namely, hemispherical block (bone), hydraulic actuator and synthetic fiber lines (muscles and tendons), silicon-based strips (ligaments), and natural rubber discs (cartilages). In order to mimic a human spine’s movement characteristics as well as its anatomy, the SJLM was assembled by concatenating the hemispherical blocks (3D-printed with ABS filament) via the four synthetic fiber lines (Dyneema Liros D-Pro SK78/2 mm/red/ 4021 N breaking strength, Liros, GmbH, Berg, Germany).

Studies in [27,34,35,36] proposed mechanical design-based approaches to constrain the wearer’s movement within a safe range. By contrast, the silicon-based strips were employed as flexible constraints for our SJLM to guarantee redundant configurations as well as the wearer’s full DoF movement. The natural rubber discs were inserted between the hemisphere blocks to serve as damper/suspension against impulsive external forces.

### 2.3. Dexterous Functionalities with Adjustable Stiffness

Figure 2 shows the stiffness adjustment mechanism of the SJLM according to hydraulic actuation. We note that the SJLM part is presented by a cross-sectional view with only two synthetic fiber lines (top and bottom) to explicitly illustrate the configuration of the SJLM when the lines are pulled/released. Recall that all hemispherical blocks and the end-link part are indeed concatenated via four synthetic fiber lines. As depicted in Figure 2a, when the lines are completely loosened by a hydraulic actuation, the concatenations of all hemispherical blocks are detached. Accordingly, both hemispherical blocks and the end-link part can move freely and the stiffness of the SJLM only depends on the rubber strips, which corresponds to the minimum stiffness. Figure 2b presents the lines as fully tightened; accordingly, the hemispherical blocks and the end-link part are aligned. In this case, the concatenation strength between the hemispherical blocks becomes almost close to the breaking strength of the synthetic fibers; consequently, The SJLM has the maximum stiffness under this configuration. If the fiber line tightness is set to in-between, then the stiffness of the SJLM is determined by a combination of the elasticity of the rubber strip and the tension of the fiber lines. In sum, the configuration as well as the stiffness of the SJLM is adjustable under our design mechanism by varying the tensile force of the synthetic fiber lines. We note that the friction between the holes of the hemisphere blocks and the synthetic fiber lines of course affects the stiffness of the SJLM; however, we mainly focus on explaining a relationship between the hydraulic actuation and the stiffness of the SJLM. The effect of this friction is modeled as a nonlinear friction term later on.

The above-mentioned adjustable stiffness furnishes the SJLM with dexterous functionalities. Figure 3 shows the weight lifting and supporting capability of the SJLM. As shown in Figure 3a through Figure 3d, the SJLM can lift up various weights ranging from 2.5 kg to 10 kg, then sustain its configuration while supporting the weight. Figure 4 demonstrates that the SJLM can play the role of an elastic link. Figure 4a through Figure 4d shows that flexion and extension up to ±90 degrees, axial torsion, flexible manipulability in 3D, and restoration after applying an external force, respectively. All these demonstrated functionalities substantiate that the SJLM can be utilized to design wearable assistive systems. For instance, suppose that the two links of the SJLM are attached to the thigh and the shin of a wearer so that its hemispherical blocks are aligned with a knee joint. By gradually tightening/loosening the line tension, the SJLM can support the wearer to perform a sit-to-stand (and the reverse, a stand-to-sit) action safely. Moreover, serious knee injuries, e.g., twist or sprain, can be prevented since the SJLM preserves the full DoF of the wearer’s knee movement. For more demonstration, please refer to Appendix A.

## 3. Modeling the Input–Output Relationship of a Spline-like Joint Link Mechanism

The relationship between a tensile force on four synthetic fiber lines under hydraulic actuation and an assistive force generated by an end-link can be modeled as input–output relationship. To derive the input–output relationship model, we employed a simplified free body diagram in which a hemispherical block and the end-link works as a line-attached lever system with a fulcrum, as illustrated in Figure 5. The effect of friction between the hemisphere blocks and the synthetic fiber lines is addressed in the last part of this derivation.

From Figure 5a, $F_{\mathrm{ten}}$ and $F_{\mathrm{gen}}$ denote the tensile force induced by tightening the synthetic fiber lines under hydraulic actuation and the generated force by the end-link, respectively. Since the end-link is of a cylindrical shape, we can assume that the direction of $F_{\mathrm{gen}}$ is passing through the center of the end-link and acting toward a radial direction. r is a vector from the center of the hemispherical block to the center of a line. A vector h points from the fulcrum point to the point where $F_{\mathrm{gen}}$ is generated. We note that, from now on, a boldface symbol represents a vector and a non-boldface symbol represents its length, i.e., ∥r∥=r.

From Figure 5b, let $d_{1},\;d_{2},\;d_{3}$ and $d_{4}$ denote vectors from the fulcrum point to the center of each line, respectively. Suppose that the four lines are tightened by $F_{\mathrm{ten}}$, then the total torque τ produced at the fulcrum point is
(1)$$\tau =d_{1}\times F_{\mathrm{ten}}+d_{2}\times F_{\mathrm{ten}}+d_{3}\times F_{\mathrm{ten}}+d_{4}\times F_{\mathrm{ten}}$$
where,
(2)$$\begin{array}{cc} d_{1} & =\sqrt{2r^{2}-2r^{2}\cos\left(\pi -\phi \right)},\\ d_{2} & =\sqrt{2r^{2}-2r^{2}\cos\left(\frac{\pi }{2}+\phi \right)},\\ d_{3} & =\sqrt{{\left(2r\right)}^{2}-{\left(d_{1}\right)}^{2}},\\ d_{4} & =\sqrt{{\left(2r\right)}^{2}-{\left(d_{2}\right)}^{2}}. \end{array}$$

This can be easily identified from geometric relationships depicted in From Figure 5b.

The torque τ can also be expressed in terms of h and $F_{\mathrm{ten}}$ as
(3)$$\tau =h\times F_{\mathrm{gen}}$$

From Equations (Equation 1) and (Equation 3), the relationship between $F_{\mathrm{ten}}$ and $F_{\mathrm{gen}}$ is
(4)$$\left(d_{1}+d_{2}+d_{3}+d_{4}\right)\times F_{\mathrm{ten}}=h\times F_{\mathrm{gen}}.$$

Due to the symmetric placement of the SJLM fiber lines, $d_{1}+d_{2}+d_{3}+d_{4}$ yields a vector whose length is always equal to 4r and whose direction is aligned with $F_{\mathrm{gen}}$ as ϕ in the *x*-*z* plane. Therefore, from Equation (Equation 4), we can obtain the input–output relationship between the amount of the tensile force, *F*_ten_ (input), and the amount of the generated force, *F*_gen_ (output), as
(5)$$F_{\mathrm{gen}}=\frac{4r}{h}F_{\mathrm{ten}}$$
which implies that *F*_gen_ linearly increases. Moreover, for all ϕ and θ, the same amount of force is generated. Lastly, we consider a friction between the four lines and hemispherical blocks as a nonlinear friction term, denoted by $\varphi (F_{\mathrm{ten}},\theta ,\phi )$, which might be expected to show a hysteresis (since the motion of the four lines passing though hemisphere block is conceptually similar to a tunneling behavior). Hence, from Equation (Equation 5), we obtain the input–output relationship
(6)$$F_{\mathrm{gen}}=\frac{4r}{h}F_{\mathrm{ten}}-\varphi \left(F_{\mathrm{ten}},\theta ,\phi \right)$$

Note that this theoretically obtained linear relationship between *F*_ten_ and *F*_gen_ is substantiated in the following section.

## 4. Experiments and Results

The following three experiments were performed to substantiate the derived input–output relationship between *F*_ten_ and *F*_gen_ (Section 4.1), identify the maximum supporting capabilities under the maximum tensile force setup which can serve as a technical specification (Section 4.2), and test the feasibility to be utilized as a key component for the design of wearable assistive devices (Section 4.3). Discussions about the results follow each experiment.

### 4.1. Substantiating the Input–Output Relationship of the SJLM

Figure 6 shows the experimental setup to substantiate the derived input–output relationship (namely, a tensile force *F*_ten_ versus a generated force *F*_gen_ relationship) which was presented in Equation (Equation 6). A desired value of *F*_ten_ could be set by using a pressure sensor (LM-PPT-H, Lumax Aerospace Co., Daejeon, Korea). Meanwhile, *F*_gen_ was measured by a push–pull gauge (FGP-50, 0.1 N resolution, maximum measured force 500 N, Nidec-Shimpo Co., Kyoto, Japan).

Recall that θ and ϕ are a leaning angle and a torsional angle of the link part of the SJLM illustrated in Figure 6, respectively. The experiment was performed by the following procedures:Install a testbed while setting the SJLM at a specific angle, e.g., $\theta =80^{\circ }$.Increase *F*_ten_ gradually by a hydraulic actuation during 60 s from 0 N to 500 N.Decrease *F*_ten_ gradually by a hydraulic actuation during 60 s from 500 N to 0 N.Measure *F*_gen_ while *F*_ten_ is being increased and decreased.Repeat the above procedures for $\theta =50^{\circ }$, $\theta =20^{\circ }$, in sequence.

Under our design, *r* and *h* were set to 11 mm and 90 mm, respectively. Figure 6b through Figure 6d presents the testbed installment for $\theta =80^{\circ }$, $\theta =50^{\circ }$, and $\theta =20^{\circ }$. ϕ determines the radial direction of the $F_{\mathrm{gen}}$ (see Figure 6). From Equation (Equation 2), we know that $d_{1}+d_{2}+d_{3}+d_{4}$ has the maximum value when $\phi =45^{\circ }$, where the fulcrum point has the maximum distance from the four holes and no hole works as the fulcrum point. To fully investigate the effect of friction between the fiber lines and the holes and model it as a nonlinear term, ϕ was set as $45^{\circ }$ for all conditions.

Figure 7 presents the substantiated input–output (a tensile force *F*_ten_ versus a generated force *F*_gen_) relationship of the SJLM. By recalling Equation (Equation 6) and substituting r=11 and h=90, we have
(7)$$\begin{array}{cc} F_{\mathrm{gen}} & =\frac{4\times 11}{90}F_{\mathrm{ten}}-\varphi \left(F_{\mathrm{ten}},\theta ,\phi \right)\\ \; & =0.489F_{\mathrm{ten}}-\varphi \left(F_{\mathrm{ten}},\theta ,\phi \right). \end{array}$$

From Figure 7, the slopes of the *F*_ten_ versus *F*_gen_ curve are approximately (at $F_{\mathrm{ten}}=300$)
$$\begin{array}{c} \frac{F_{\mathrm{gen}}}{F_{\mathrm{ten}}}≃\frac{130}{300}=0.433\;\mathrm{when}\;\;\theta =80^{\circ }\\ \frac{F_{\mathrm{gen}}}{F_{\mathrm{ten}}}≃\frac{120}{300}≃0.400\;\mathrm{when}\;\;\theta =50^{\circ }\\ \frac{F_{\mathrm{gen}}}{F_{\mathrm{ten}}}≃\frac{110}{300}≃0.367\;\mathrm{when}\;\;\theta =20^{\circ } \end{array}$$
which allows us to presumably estimate the effect of a frictional force $\varphi (F_{\mathrm{ten}},\theta ,\phi )$ from Equation (Equation 7) as follows:$$\begin{array}{c} \varphi \left(F_{\mathrm{ten}},80^{\circ },45^{\circ }\right)≃0.056F_{\mathrm{ten}}\;\mathrm{when}\;\;\theta =80^{\circ }\\ \varphi \left(F_{\mathrm{ten}},50^{\circ },45^{\circ }\right)≃0.089F_{\mathrm{ten}}\;\mathrm{when}\;\;\theta =50^{\circ }\\ \varphi \left(F_{\mathrm{ten}},20^{\circ },45^{\circ }\right)≃0.122F_{\mathrm{ten}}\;\mathrm{when}\;\;\theta =20^{\circ } \end{array}$$

However, since the *F*_ten_ versus *F*_gen_ curve shows an obvious nonlinear hysteresis, $\varphi (F_{\mathrm{ten}},\theta ,\phi )$ needs to be identified further. Compared to our results, similar hysteresis plots were introduced in [37] as well. The nonlinear hysteresis would be approximated by adopting a neural net-based approach.

Suppose that h=157 mm, which is a half of the shank length as presented in [38]. We then calculate the maximum assistive torque $\tau_{\max}$ from Equation (Equation 3) ($\theta =80^{\circ }$ is assumed)
(8)$$\begin{array}{cc} \tau_{\max} & =hF_{\mathrm{gen}}=h\frac{F_{\mathrm{gen}}}{F_{\mathrm{ten}}}F_{\mathrm{ten}}\\ \; & =0.157\left[m\right]\times 0.433\times 3270.194\left[N\right]\\ \; & =219.999\left[\mathrm{Nm}\right]≃220\left[\mathrm{Nm}\right] \end{array}$$
where 3270.194[N] corresponds to 80% of the maximum breaking tolerance of the synthetic fiber, which guarantees the safe operation of the SJLM experimentally. The outcome is competitive with those for the existing lower-limb exoskeletons, such as those presented in [14,18,21], of which the maximum assistance force ranges from 50.0 Nm to 200.0 Nm.

### 4.2. The Maximum Supporting Capabilities of the SJLM under the Maximum Tensile Force

Figure 8 shows the experimental setup to measure the maximum supporting capabilities of the SJLM under the maximum tensile force, which can serve as a technical specification of the SJLM. An absolute Digimatic indicator (ID-S112XB, 0.001 mm resolution, Mitutoyo Co., Kawasaki, Japan) and a push–pull gauge (FGP-50, 0.1 N resolution, maximum measured force 500 N, Nidec-Shimpo Co.) were used to measure the applied force and corresponding displacement, respectively. The goal of this experiment was to identify the maximum supporting capabilities of the SJLM in terms of the maximum translation and rotational stiffness. The experiment was performed as follows:Apply the maximum tensile force $F_{\mathrm{ten}}$ to the SJLM.Push the SJLM upward by the push–pull gauge and measure the force applied to the SJLM.Meanwhile, measure the displacement of the SJLM by the absolute Digimatic indicator (translational displacement).Repeat the above procedures ten times (the applied $F_{\mathrm{ten}}$ is supposed to be above 196 N).Convert the measured translational stiffness into the rotational stiffness, and calculate the mean and standard deviation.Calculate the translational/rotational stiffness based on the measured translational/rotational displacement.

From the experiment, the measured translational displacement was 0.567 ± 0.039 mm, which corresponded to the rotational displacement (6.365 ± 0.436) × 10${}^{-3}$ rad according to a torque arm of 89 mm when the applied force was 213.96 ± 12.19 N, in terms of torque 19.04 ± 1.09 Nm. This results indicated that, under the maximum tensile force setting, the maximum translational and rotational stiffness of the SJLM were 378.1 ± 9.6 kN/m and (2.995 ± 0.076) × 10${}^{3}$ Nm/rad, respectively. These resulting values are competitive with the joint stiffness of the Dexter arm [39,40] consisting of gearbox, motor, and steel wire, which is commonly used for exoskeletons [13,22,23,24,25,26].

### 4.3. Feasible SJLM Applications: Wearable Lower-Limb and Spine Assistive Devices

Figure 9 shows the two feasible applications of the SJLM. First, a wearable lower-limb assistive device is presented in Figure 9a. Two SJLMs are mounted on the lateral sides of the pilot wearer’s legs; specifically, the hemisphere block parts are aligned with the wearer’s knee joints. If the SJLM were set to a high stiffness adjustment, then the pilot wearer could maintain the upright position for a longer time but would expend less musculo-skeletal power. Figure 9b demonstrates that the SJLM can be bent with elasticity while the DoF for the wearer’s movement is preserved. Next, Figure 9c presents a spine assistive device designed as a harness type. From Figure 9d, we can see that the pilot wearer’s spine (especially a thoracic/lumbar spine) can be supported by the SJLM. This type of wearable assistive device will be helpful for workers who need to sustain an upright posture for a long time, e.g., mart cashiers, security guards, and so on.

Recall that the design of the SJLM is mainly aimed at providing the capability for adjusting its stiffness/flexibility to assist a wearer. We expect that this capability enables the SJLM to be utilized as a key component to develop whole-body wearable assistive devices with the following advantages:*Under high-stiffness (low-flexibility) adjustment, the SJLM can support the wearer with the desired assistive force*: This advantage is well-exemplified by the existing Chair Exoskeleton (CEX) of Hyundai Motor Group [41] (see Figure 10a).*Under low-stiffness (high-flexibility) adjustment, the SJLM can provide the wearer with comfort by alleviating kinematic discrepancy and joint misalignment*: As presented in [42], for instance, the HeroWear Apex suit—a biomechanically assistive garment to support the lumbar spine—is light and comfortable to wear (see Figure 10b).

## 5. Conclusions

This paper proposed a spine-like joint link mechanism (SJLM) for designing wearable assistive devices with comport and support, which guarantees better mobility. Inspired by a human spine structure, the SJLM was made of the concatenation of multiple hemispherical blocks via four synthetic fiber lines. The concatenation stiffness of the hemispherical blocks could be adjusted by regulating the tensile force on the lines under a hydraulic actuation, which in turn generated a supporting force at the end-link of the SJLM. The experimental results substantiated the feasibility of the SJLM to be utilized as a key component for designing wearable assistive devices by identifying input–output characteristics as well as the maximum supporting capabilities under the maximum tensile force setup. Two SJLM applications were presented to demonstrate the substantiated feasibility: these were wearable lower-limb and spine assistive devices.

We conclude this paper by providing additional information about the SJLM from a research as well as a market perspective. First, the SJLM was designed to minimize the risk of the wearer’s injuries, e.g., joint sprain or dislocation, when an unexpected fall occurs. Nonetheless, the wearer should be aware of the potential failure of the SJLM which might be induced by a hydraulic piston abrasion, a sudden power cut, and so on. Next, as aforementioned, the specification of the SJLM can be flexibly modified according to the wearer’s body measurement and design materials. Therefore, the cost affordability can also be secured by the designer’s choice.

This study will culminate in the design of an innovative version of wearable assistive devices, such as a lower-limb exoskeleton or an upper-limb augmentation device, in future works.

## Figures and Tables

**Figure 1 sensors-22-02314-f001:**
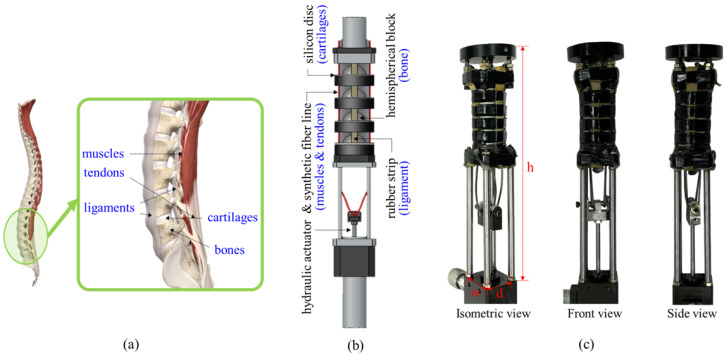
The SJLM design inspired by a human spine and functional correspondences: (**a**) human spine anatomy (excerpted and redrawn from https://3d4medical.com (accessed on 23 October 2021)), (**b**) SJLM components, (**c**) the fabricated SJLM: isometric view, front view, and side view.

**Figure 2 sensors-22-02314-f002:**
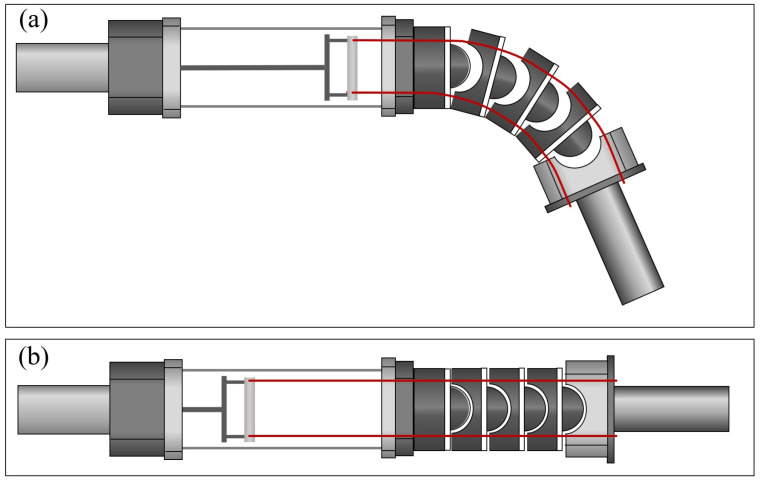
The stiffness adjustment mechanism of the SJLM according to hydraulic actuation: (**a**) when the tension of synthetic fiber lines is fully loosened and (**b**) when the tension of synthetic fiber lines is fully tightened.

**Figure 3 sensors-22-02314-f003:**
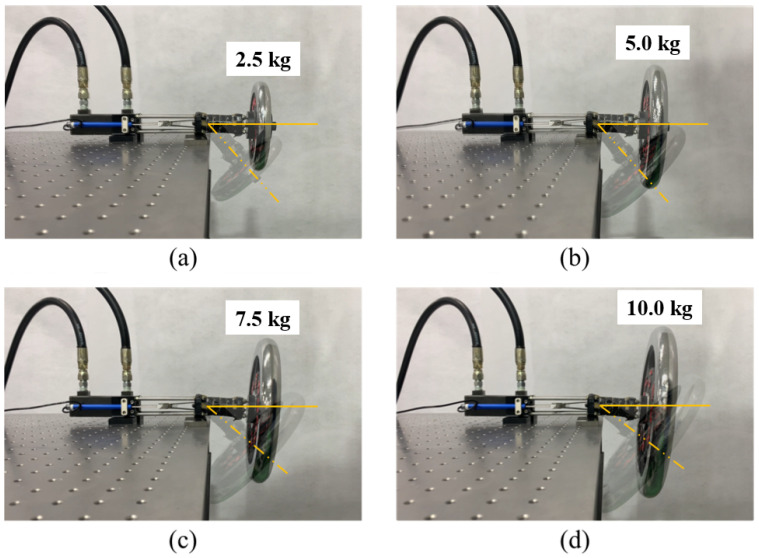
The weight lifting and supporting capability of the SJLM (the transparent areas represent initial positions and the opaque areas are final positions): (**a**) 2.5 kg weight, (**b**) 5.0 kg weight, (**c**) 7.5 kg weight, (**d**) 10.0 kg weight.

**Figure 4 sensors-22-02314-f004:**
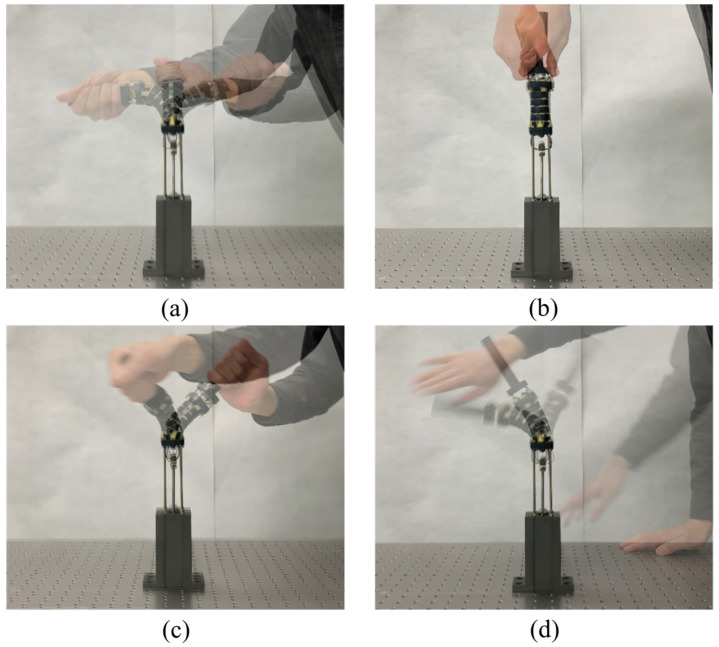
The demonstration of dexterous functionalities of the SJLM: (**a**) flexion and extension up to ±90 degrees, (**b**) axial torsion, (**c**) flexible manipulability in 3D, and (**d**) restoration after applying an external force.

**Figure 5 sensors-22-02314-f005:**
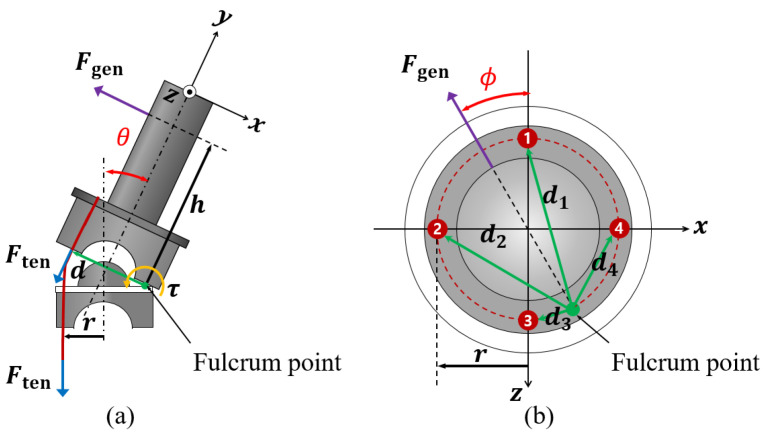
Employed lever system model to analyze the input–output relationship between $F_{\mathrm{ten}}$ and $F_{\mathrm{gen}}$: (**a**) the side view (*x*-*y* plane) of the last hemispherical block and the link and (**b**) the top view (*x*-*z* plane) of the last hemispherical block.

**Figure 6 sensors-22-02314-f006:**
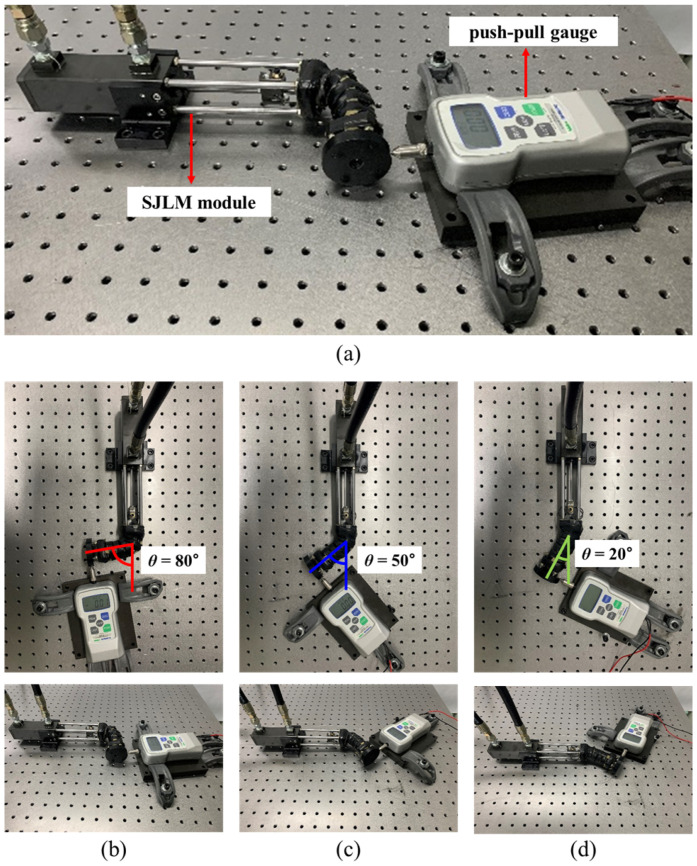
The experimental setup to substantiate the input(*F*_ten_)–output(*F*_gen_) relationship: (**a**) the testbed consists of the SJLM module and a push–pull gauge; (**b**–**d**) are experimental setups for $\theta =80^{\circ }$, $\theta =50^{\circ }$, and $\theta =20^{\circ }$, respectively. ϕ was set to $45^{\circ }$ for all conditions.

**Figure 7 sensors-22-02314-f007:**
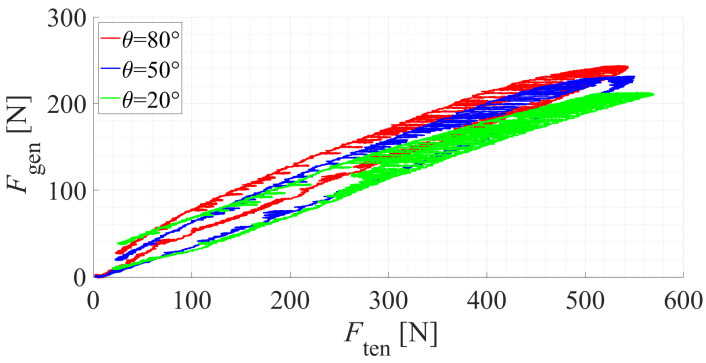
The substantiated input–output (a tensile force *F*_ten_ versus a generated force *F*_gen_) relationship of the SJLM.

**Figure 8 sensors-22-02314-f008:**
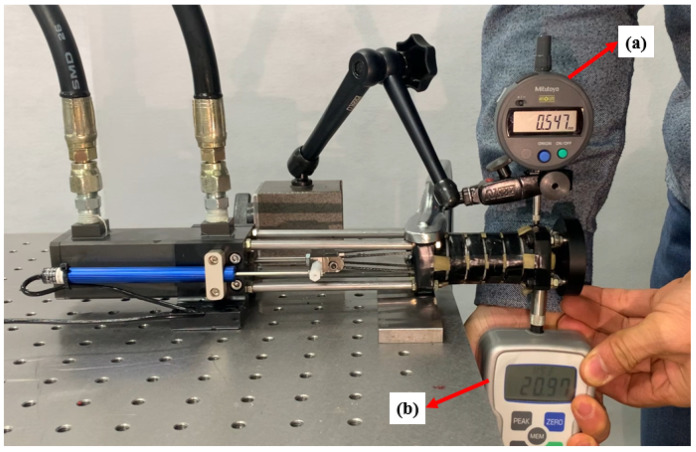
The experimental setup to measure various SJLM properties under the maximum tensile force: (**a**) an absolute Digimatic indicator to measure a displacement and (**b**) a push–pull gauge to measure a force applied to the SJLM.

**Figure 9 sensors-22-02314-f009:**
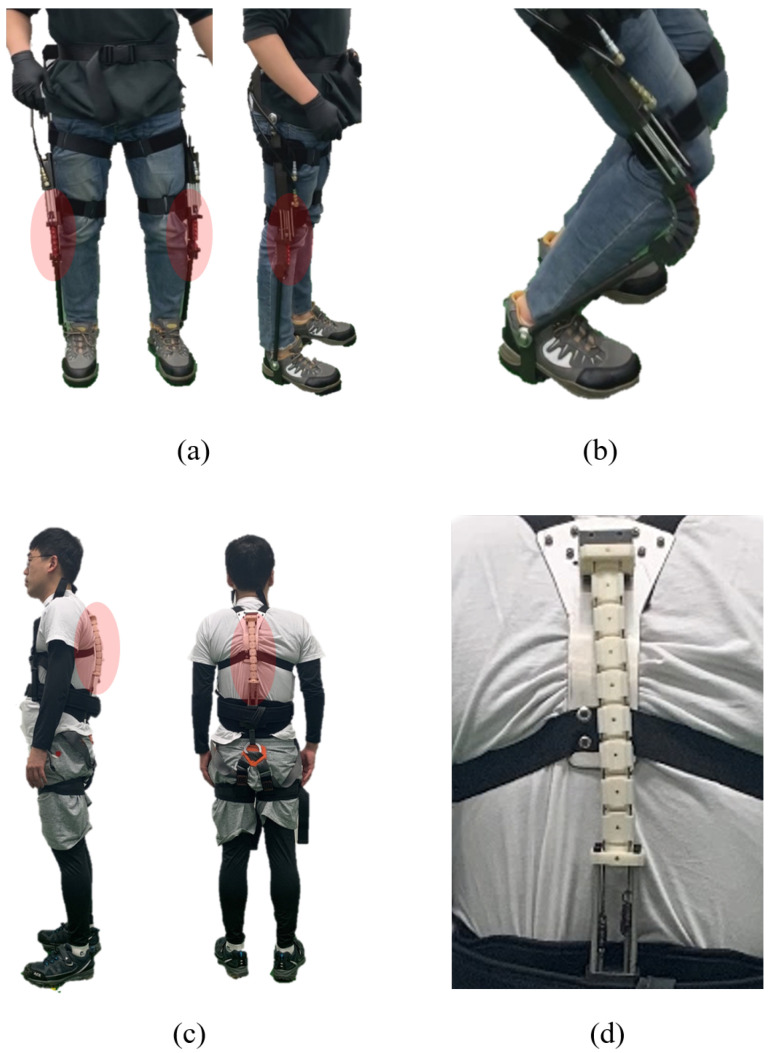
The two SJLM applications: (**a**) a lower-limb assistive device under high-stiffness adjustment and (**b**) bent with elasticity under low-stiffness adjustment; (**c**,**d**) a spine assistive device designed as a harness type to support a thoracic/lumbar spine.

**Figure 10 sensors-22-02314-f010:**
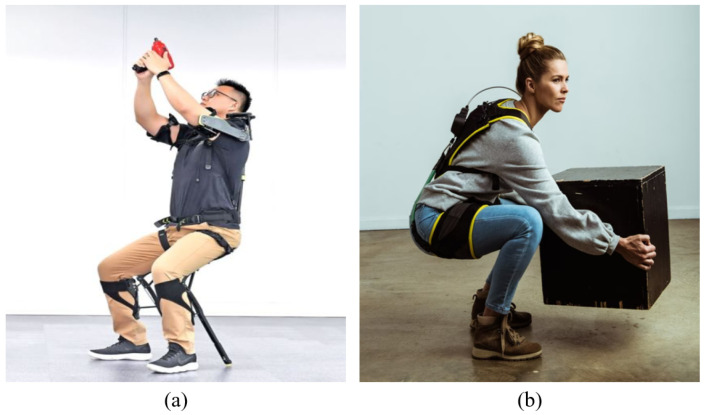
(**a**) Chair Exoskeleton (CEX) developed by Huyndai Motor Group [41]; (**b**) HeroWear Apex suit developed by HeroWear LLC [42].

**Table 1 sensors-22-02314-t001:** The specification of the SJLM and its hydraulic cylinder.

	Specification	Value	Unit	Remarks
SJLM	Size (w × d × h)	36×36×190	mm	without cylinder
	Weight	0.250	kg	without cylinder
Hydraulic Cylinder	Stroke	60	mm	-
	Piston radius	25	mm	-
	Rod radius	4	mm	-
	Weight	0.450	kg	-
	Maximum force	3347	N	at 70 bar

## Data Availability

Not applicable.

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
