# Peer review of "Spine-like Joint Link Mechanism to Design Wearable Assistive Devices"

_sensors, 2022, doi:10.3390/s22062314_

Round 1

Reviewer 1 Report

The authors presented a specific as they call-it spine-like mechanism suitable for wearable assistive devices. The intro, the background and the mechanics themselves are adequately presented which also applies to the two application modes included. There remain a few comments, only.

Page 2, line 60-62: hard to follow and understand what is meant here by 1000 percent. Please clarify.

Figure 2b: please add dimensions and sizes.

Regarding the mechanism itself is it limited regarding to various sizes due to the concept and the chosen materials or can it be miniaturized, as well?

Figure 3 legend: please omit one "the" here

Conclusion: could you gave any hint about the costs of your construct? Are there any systems on the market to compare with? What about potential failure modes regarding the mechanism?

Author Response

Please refer to the attached file. Thanks for your valuable comments.

Reviewer 2 Report

This manuscript proposes a novel joint link mechanism inspired by a human spine structure as well as functionalities. The work is well designed, and the logical is well present. In my opinion, this manuscript is suitable for the journal, and deserves a publication after addressing the following issues.

  1. The introduction is too long. It is suggested to be simplified to make the article more logical and readable.
  2. In the experimental part, whether multiple measurements were carried out. If any, it is recommended to add deviation.

Author Response

(The authors gave the same response as above.)

Reviewer 3 Report

The primary objective of this study was to design, develop, and test a novel joint link mechanism inspired by the human spine structure and functionalities. The authors present a good background highlighting the limitations and problems of the current designs of the rigid/soft-type wearable robots, especially addressing the problem of comfort and support that are widely criticized in existing wearable assistive and rehabilitative technologies. The authors talked about the conventional approach to designing wearable technologies and how it focused on simplifying the human joint-link mechanism and how this approach has resulted in discomfort and/or inadequate support and increases risk of joint or ligament injury in users. The last paragraph in the introduction section is well written and provides a good overview and structure of the manuscript to the readers. The design and model of the SJLM prototype was clearly presented.

Here are some comments I have:

1) This manuscript focuses on the design and development of new device and although the authors provided sufficient information to show the need for their novel solution, it would be good for them to mention the long-term goal for developing the new joint-link mechanism as an objective to "Design and test feasibility/efficacy of a novel joint-link mechanism mimicking the human spine.." or something along these lines to identify/clarify the primary purpose of the study to all the readers from different backgrounds.

2) Image 2(c) can be improved by providing a more isometric/dimetric view of the device instead of the current view as it is difficult to really understand what the device looks like. The authors did a good job of providing the details of the designed components and the part of the spine they were analogous to but, it would also help for figures 2(a) and 2(b) to be side by side in the same orientation for a better understanding of the design. It may be helpful to use one image with the spine on one side and the SLJM on the other side and label the 4 different components for both (the spine as well as the SJLM) for readers to gain a better insight into the design. It would also help if the authors provided a couple of views to show the 4 different strings being used to concatenate the hemispherical blocks.

3) The authors also talk about using hydraulic actuation to tighten or loosen the strings and how the stiffness can be adjustable but, there is no clear information on how this stiffness can be varied. The authors should elaborate on this a little bit more as it is one of the key components of their design to ensure a more natural degree of freedom.

4)  The authors followed a standard procedure for testing the dexterous functionality by subjecting the SJLM to bending loads (discussed in Section 2.3 and shown in Figure 4). did the authors also perform any compressive loading/tensile loading tests? If so, could they share some information regarding this as well?

5) The modeling of the input-output relationship is well done but, it would be helpful if the authors clarified the definitions of the angles theta and phi. From what I see, phi is a torsional angle, which can be analogous to internal/external rotation in the spine. Why is this angle set to 45 in the experimental setup? And how is this angle set at 45? There is no clear explanation about how this was done.

6) The weight and dimensions of the SJLM were not presented. The specification for the hydraulic actuator was not provided, which directly associated with the wearability of the system should be added.

7) It would help if the authors presented a list of steps for both testing protocols with a few more details mentioned above to ensure repeatability of experiment.

8) The main research question was not completely addressed because the authors mention the improved comfort and support with this design but there was no data to prove the improved comfort or support using this new joint link mechanism. The authors should modify title to not include the comfort and support in it and focus on talking about the potential improvement in comfort and support in the "discussion" section or the "feasible applications section". Currently, the manuscript focuses on bench-testing the new joint link mechanism design and showing proof between the theoretical calculations/derivations.

9) The safety was mentioned in the abstract and conclusion, but no data in the results or discussion to support the safety of the proposed device. Actually I do have concerns on the safety as mechanically there does not seem any protection existed.

Author Response

(The authors gave the same response as above.)

Round 2

Reviewer 3 Report

Thank you for addressing my comments thoroughly. I am excited about the technology and looking forward to reading more user evaluations and applications. I would like to suggest you include people with disabilities and therapists in your next work to maximize your success.